# Family lexicon: Using language models to encode memories of personally familiar and famous people and places in the brain

Andrea Bruera[1,2]*, Massimo Poesio[1]

**1** Max Planck Institute for Human Cognitive and Brain Sciences, Cognition and Plasticity Research Group, Leipzig, Germany, **2** Queen Mary University of London, London, United Kingdom

* bruera@cbs.mpg.de

**Data Availability Statement:** Because of privacy reasons, it is not possible to publicly share the responses to the questionnaires, nor the subject-specific computational representations extracted from them. The parts of the data that could be

## Abstract

Knowledge about personally familiar people and places is extremely rich and varied, involving pieces of semantic information connected in unpredictable ways through past autobiographical memories. In this work, we investigate whether we can capture brain processing of personally familiar people and places using subject-specific memories, after transforming them into vectorial semantic representations using language models. First, we asked participants to provide us with the names of the closest people and places in their lives. Then we collected open-ended answers to a questionnaire, aimed at capturing various facets of declarative knowledge. We collected EEG data from the same participants while they were reading the names and subsequently mentally visualizing their referents. As a control set of stimuli, we also recorded evoked responses to a matched set of famous people and places. We then created original semantic representations for the individual entities using language models. For personally familiar entities, we used the text of the answers to the questionnaire. For famous entities, we employed their Wikipedia page, which reflects shared declarative knowledge about them. Through whole-scalp time-resolved and searchlight encoding analyses, we found that we could capture how the brain processes one's closest people and places using person-specific answers to questionnaires, as well as famous entities. Overall encoding performance was significant in a large time window (200-800ms). Using spatio-temporal EEG searchlight, we found that we could predict brain responses significantly better than chance earlier (200-500ms) in bilateral temporo-parietal electrodes and later (500-700ms) in frontal and posterior central electrodes. We also found that XLM, a contextualized (or large) language model, provided superior encoding scores when compared with a simpler static language model as word2vec. Overall, these results indicate that language models can capture subject-specific semantic representations as they are processed in the human brain, by exploiting small-scale distributional lexical data.

## 1 Introduction

Being asked to describe one's closest friend, or one's favourite neighbourhood, is not an easy question to answer. One will find that just describing physical and personality traits is not

published, together with the code, are publicly available online, on the Open Science Foundation website (https://osf.io/sjtmn).

**Funding:** AB was supported by a doctoral studentship from the School of Electronic Engineering and Computer Science, Queen Mary University of London. The funders had no role in study design, data collection and analysis, decision to publish, or preparation of the manuscript.

**Competing interests:** The authors have declared that no competing interests exist.

enough—to do justice to that specific person or place it will be necessary to bring up much more. Anecdotes, past events and stories involving them, together with other disparate pieces of information that just come up to one's mind when talking about familiar entities. And, taken together, this will form an extremely idiosyncratic mixture, that however captures the fundamental uniqueness of that person or place in one's memory.

In cognitive neuroscience, as a reflection of this, it has been found that knowledge for individual entities, such as people and places, is particularly rich and multifaceted [1–3]. Aside from its episodic (event-specific) and semantic (encyclopedic) components [4–6], it seems to strongly involve another type of knowledge, called 'personal semantics' [7]. Personal semantics can be described as a type of knowledge which is abstracted from individual events and occurrences, but not fully so. An example could be knowing what a friend enjoys doing. It is not entirely encyclopedic, as it is dependent from memories of repeated events that took place in the past. Nor is it completely episodic, as it does not strictly depend from individual instances of that event. In this sense, it is part of personal semantics.

Episodic, semantic and personal knowledge compose declarative, or explicit, memory [6, 7]. By definition, declarative memory is knowledge that can be expressed through natural language [8]. For generic concepts, such as a cello, declarative knowledge is widely shared across a community of speakers. Because of this, it can be easily extracted in a data-driven fashion from large collections of text (called **corpora**) such as Wikipedia. This is done by creating vectorial representations, reflecting distributional properties of their corresponding words in the corpora, through **language models**—also called **distributional semantics models**. Such approaches follow the hunch that the semantic content of a word can be captured by the way in which this word is used [9]. To take a simplified example, an important part of the meaning of the word 'leaf' can be captured by observing that it co-occurs frequently in natural language with words like 'tree', 'branch' or 'flower', but much more rarely with words such as 'citizen', 'musician' or 'factory'. This, in turn, can be operationalized as the so-called distributional hypothesis—that words having similar meanings will be found in similar linguistic contexts [10]. The resulting representations are traditionally called **word vectors**. Word vectors have been shown to capture and encompass in a single high-dimensional vectorial space multiple traditional semantic dimensions proposed in cognitive science [11–14]. This can explain their effectiveness at modelling semantic processing, both in behaviour [15, 16] and in the brain [17–20].

However, when it comes to personally familiar people or places, two main challenges arise. First, the extraction of semantic representations for concepts based on distributional information requires words to be frequent enough in the corpora in order to robustly capture their meaning [21–25]. Despite their fundamental importance in our lives, personally familiar people and places (a close friend, one's favourite neighbourhood) never—or extremely rarely—get mentioned in large-scale corpora such as Wikipedia. Therefore, it seems impossible, in principle, to capture their meaning by way of word vectors.

Secondly, even if one were able to sidestep this issue, a more pervasive one would emerge: namely, that each person has highly idiosyncratic and subjective ways of perceiving and describing personally familiar people and places. This makes it hard to capture semantic representations from recollections of autobiographic memories expressed in natural language, which constitute an exceptionally diverse and reduced linguistic dataset [26–28].

Such limitations posed by language models have had an impact on studies employing them as models to capture semantic representations in the brain. As a consequence of the need for sufficient training data, previous work has focused on generic concepts for whom a representation could be obtained from large corpora [17, 18, 29, 30]—or, in the case of individual entities, on famous entities which are mentioned with enough frequency in corpora [31]. The only

partial exception taking into consideration subject-specific semantic knowledge is, to our knowledge, [32], which, however, focused on personal interpretations of generic concepts (e.g. personal interpretations of 'dance' as an event) and not of individual entities. Previous work in cognitive neuroscience looking at subject-specific knowledge of individual entities has not used distributional linguistic information, but rather semantic dimensions defined *a priori* by the experimenters [33–36].

In this work, we set out to investigate whether we could use short texts containing personal memories to build unique vectorial semantic representations for personally familiar entities, such as people and places, that could encode the way in which the brain of each subject processes such entities. Our approach follows a framework that has been recently emerging in neuroscience, aiming at recognizing and effectively accounting for the uniqueness of individuals and of their cognitive and neural processes [32, 37–40].

In our experiment, first we captured person-specific knowledge in the form of text. We did so by asking subjects to talk about the most important people and places in their lives (eight people and eight places; see Fig 1). This allowed us to collect textual data regarding semantic knowledge (physical and personality traits), episodic memory (the most salient memories involving a person/place) and personal semantics knowledge (operationalized as words and topics associated with each person/place). We then used language models to encode each subject's text, thus creating subject-specific vector representations of personally familiar entities from small-scale textual distributional information. We also created in a mirrored way vector representations for eight famous people and eight famous places, as a 'control' set of entities, for which it is known that language models are able to create reliable semantic representations given their high frequency in corpora [31, 41, 42].

In parallel, we collected electroencephalography (EEG) data while the same subjects were reading the entities' names (both famous and familiar). We ran a set of encoding analyses, looking at where in time (time-resolved encoding using all electrodes) and in space (spatio-temporal searchlight encoding, looking at clusters of electrodes separately) our subject-specific vector representations captured brain processing. Names were used instead of faces in order to avoid non-semantic, low-level visual differences among different categories of stimuli (people and places, famous and personally familiar).

We compared two language models for Italian, the language in which the experiment was carried out, from two different families of language models. The first model is **word2vec** [43], a so-called static model. In static models, each word is represented as a single, fixed vector representation. This captures distributional information encoded in the training data only for word types (i.e. all aggregated mentions of each word), but not tokens (i.e. actual individual occurrences in context). Static models are the most commonly used models in brain encoding/ decoding studies that involve individual concepts [18, 20, 29, 30, 44]. The second model is XLM-Roberta-large (**XLM**, [45]), a so-called **contextualized language model**, or **large language model** (**LLM**). LLMs are more recent and much more complex than static models, and they are based on the Transformer architecture [46]. They do not represent word *types* (as static models do), but word *tokens*: given a linguistic context, such as a sentence, the vectors for the words are created by adapting pre-trained representations to the specific sentence— thus incorporating both type- and token- level information. LLMs have been used to encode or decode to/from the brain the meaning of words in context—i.e. words appearing in passages of text of various complexity, ranging from phrases [47] to sentences [48] and narratives [49– 51]. However, this type of model has never been tested, to our knowledge, as a model for personal, subject-specific semantic representations of extremely specific entities like people and places.

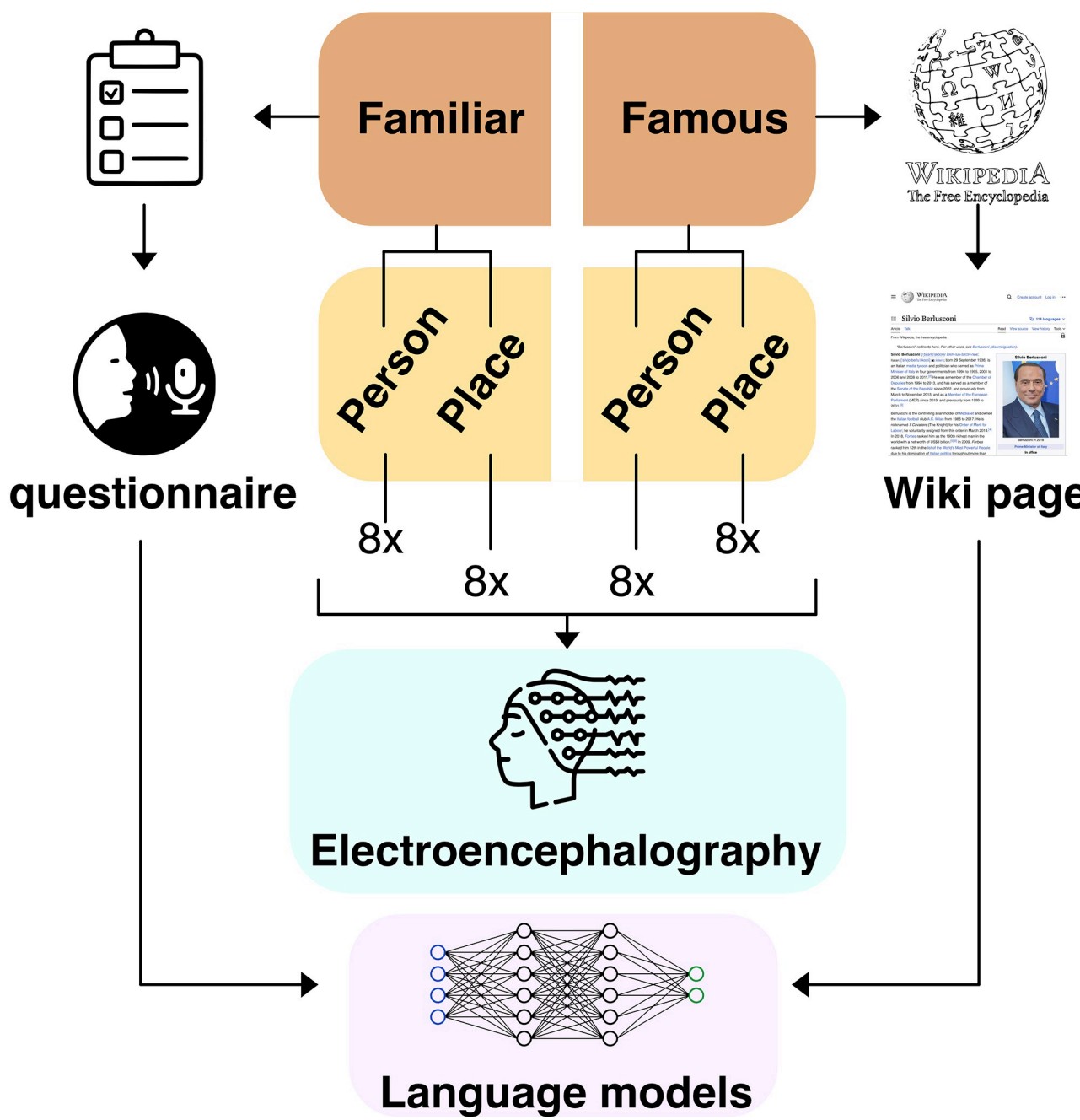

**Fig 1. Visualization of the experimental setup.** Our stimuli were divided into proper and personally familiar names. For each level of familiarity, sixteen famous individual entities (eight people and eight places) were present. Famous entities were selected after controlling for familiarity, imageability and name length. Personally familiar names were obtained by asking each participant to provide the names themselves. We also collected short amounts of text describing each entity (Wikipedia pages for famous entities, and answers to a questionnaire for personally familiar people and places). From these, we extracted one semantic vector per entity using a language model. Then we collected the EEG data, and carried out separately for each participant the encoding analyses.

The results reported in Figs 3 and 4 indicate that, by encoding personal memories with language models, it was possible to create semantic representations of individuals that captured the way in which they are processed in the brain. Importantly, we were able to create semantic representations for the most important people and places in each subject's life using only the

reduced amount of text available to us through a questionnaire –i.e. small-scale distributional lexical information. Furthermore, this approach worked also for a matched set of famous entities, for which non-personal textual information obtained from Wikipedia was used.

## 2 Methods

### 2.1 Stimuli

**2.1.1 Famous entities.** As one half of the experimental stimuli, we selected eight famous people and eight famous places to be used for all participants (see Fig 1, right portion). Before each EEG experimental session took place, we made sure that subjects had neither personally known any famous person nor visited a famous place among the stimuli. Only two subjects had visited one of the famous places selected as experimental stimuli. In those cases we substituted the names with other place names matched for length, familiarity and imageability. All ratings, including those for the famous places that acted as substitutions, can be found online together with the code and the data (see below).

We balanced the stimuli in terms of name length, familiarity and imageability, in order to avoid significant differences across the two semantic categories. The final set of stimuli was selected from a larger set of 100 stimuli, for which we obtained familiarity and imageability ratings. The subjects for the rating tasks (carried out separately; 33 subjects for familiarity, 30 for imageability) were native speakers of Italian, and none of them took part to the EEG experiment. Familiarity was defined in the same way as in [52]—a quantification on a scale from 1 to 5 of the number of cumulative encounters with an individual entity, across time and media. The average familiarity overall was 3.59 (standard deviation 0.44); average familiarity for people was 3.75 (standard deviation 0.45), while that for places was 3.44 (standard deviation 0.36), and their difference across people and places was not statistically significant (non-parametric two-sample permutation test: $t = 1.42$, $p = 0.15$). Imageability was defined after [53] as the ease or difficulty of arousing a mental image. Imageability was controlled because, during the experiment, subjects were asked to read names and picture their referents mentally (see Section 2.2). We used the most common scale in imageability rating experiments, going from 1 to 7 [54]. The average imageability across all entities was 4.97 (standard deviation 0.85). Average imageability for people was 5.17 (standard deviation 0.66), while average imageability for places was 4.78 (standard deviation 0.96). The difference between the two was not statistically significant (non-parametric two-sample permutation test: $t = 0.88$, $p = 0.37$). Name length was on average overall 13.5 (standard deviation 3.12); average person name length was 14.1 (standard deviation 1.83), while average place name length was 13 (standard deviation 3.9), and the difference across categories was again not significant (non-parametric two-sample permutation test: $t = 0.68$, $p = 0.47$). All ratings are provided together with the code for replication (see below).

For each famous individual entity we also collected the text from their Wikipedia page, under the assumption that such texts are a source of explicit knowledge regarding individual entities that can be mapped to the brain using their distributional information—an approach validated in neuroscience in [55] and in Natural Language Processing (NLP) in [41, 56, 57]. These texts will be used as described in Section 2.4.1 in order to extract semantic representations using language models. We also manually annotated for each famous person their occupation (e.g. politician, musician) and for each famous place its type (e.g. city, monument), since this information was used during the experimental task (see Section 2.2).

**2.1.2 Personally familiar entities.** Before starting the EEG experiment, we asked participants to provide the names for eight personally familiar people and eight places (see Fig 1, left portion). As a framework, we followed research on social circles [58]. For people we focused,

in our definition, on members of the so-called 'support clique'. This circle consists of people with whom one has a positive relationship, is in touch regularly, and from whom one would seek personal advice or help [59]. For places, we tried to match as closely as possible the definition given for people, since no relevant literature was available. We defined 'support places' as places with whom one has a special, positive relationship, and they are places where one would (if possible) return to when in a situation of distress. We provide as S1 File the text of the specific instructions given to the subjects, translated to English (see S1 File).

Additionally, subjects were asked to respond to a questionnaire, whose aim was capturing the main components of declarative knowledge about personally familiar entities—i.e. what is explicitly known about them. We looked at the two components most traditionally associated with explicit knowledge (semantic memory and episodic memory, which differ by being respectively dependent or not from specific events [4, 6]), as well as at what has been called personal semantics [7], which is a highly personal type of knowledge, lying at the intersection of episodic and semantic memory (see Introduction). The questionnaire therefore involved nine questions, divided equally among the three types of knowledge. For semantic memory, we asked to talk about the type of relationship being shared, and physical and personality traits; for episodic memory, we asked to talk about how each entity was first encountered, as well as two most recent salient autobiographical episodes involving that entity; for personal semantics, we asked to name up to 10 words that came to mind when thinking about that entity, what one would talk about (for people) or do (for places) if they met or visited that entity, and finally a sentence that is associated with a person/place. Notice that participants were not only asked to provide names and answers to the questionnaires, but also to provide either the person's occupation or the type of place (i.e. monument, city, river, etc.), to be used during the experimental paradigm (see Section 2.2).

We recorded the answers using a Zoom H2 stereo digital audio recorder, and we automatically transcribed the texts using Microsoft Azure's Speech-To-Text service. Before extracting the semantic representations from the texts using the language models (see Section 2.4), we checked the automatic transcriptions in order to verify that quality was sufficiently good, which we found to be the case.

In the case of personally familiar names, we could not control in advance for name length, since participants came up with the names. Therefore, we implemented in the analyses a procedure for explicitly removing all variance associated with name length from the EEG data, described below in Section 2.5.1.

Notice that, because of privacy reasons, it is not possible to publicly share the responses to the questionnaires, nor the subject-specific computational representations extracted from them. The parts of the data that could be published, together with the code, the ratings and all results and plots, are publicly available online, on the Open Science Foundation website (https://osf.io/sjtmn/). Also, notice that the same dataset has been used for a different study, focused on decoding semantic categories [60].

## 2.2 Experimental paradigm

Thirty-tree right-handed subjects (age from 20 to 31 years old, with 21 female participants) took part to the experiment. Sample size was determined following [61], where the authors show thirty-two subjects is an adequate sample size for event-related potentials (**ERP**) studies involving semantic processes. As the experiment was conducted in Italian, all the subjects were native Italian speakers. All experimental procedures were approved by the Ethical Committee of SISSA, Trieste, where the data were collected. The recruitment of subjects and data

collection took place concurrently between June and September 2021. Before the experiment took place, subjects gave their written informed consent.

Before the EEG experiment subjects provided names for personally familiar stimuli, as well their occupations or types of places. Then, participants took part to 24 experimental EEG runs. Each name would appear once per run, in randomized order. For each name we thus recorded twenty-four ERPs, which were averaged after preprocessing and before entering the analyses. This is routinely done in encoding/decoding studies for EEG in order to improve the signal-to-noise ratio [62].

Each trial proceeded as follows. First, a fixation cross appeared for 500 ms. Then a name appeared on screen for 500 milliseconds, followed by a fixation cross lasting 1 second. Subjects were instructed to read the name and mentally picture its referent. This was done because, even if the mental imagery task may bias participants to focus on visual semantic features, it has been shown to elicit solid semantic processing and to provide good performance in encoding/decoding [18, 31, 63]. After the fixation cross disappeared, a binary yes/no question appeared on screen. The question was added exclusively to ensure participants actually engaged in the task. To avoid strategic preparation, questions were randomized among two templates. They could reflect either a coarse-category question (e.g. 'is it a person/place?'), or a fine-grained question (e.g. 'does the name refer to a student?' or 'does the name refer to a city?'), a methodology previously used in [31, 64]. Questions were balanced between yes/no answers, and subjects could answer using two keys. In the case of fine-grained questions, also the occupation or place type was randomized.

## 2.3 EEG data collection and preprocessing

The EEG data was collected using a BIOSEMI ActiveTwo system with 128 channels, recording unfiltered signals at a sampling rate of 2048 hz. We also collected signals from two electro-oculogram channels (EOG) so as to be able to use them later for artifact rejection with Indipendent Component Analysis (ICA; details below). For preprocessing, we adapted an automated procedure using MNE (originally standing for Minimum Norm Estimate) [65], previously validated in [66].

We set the montage to the default for the 128-channel BIOSEMI system. A visualization of the montage, together with the codes for the channels, is included in the publicly available code and can be retrieved online (https://www.biosemi.com/pics/cap_128_layout_medium.jpg). Then, following [66], we used the standard ICA-based ocular artifact removal implemented in MNE. We applied a low-pass filter to the ERP data to 80 hz [67]. Then we epoched the data and subsampled it to 200hz, and removed the independent components correlating the most with eye-movement artifacts [66]. Baseline correction consisted of the average signal between 100 and 0ms before the appearance of a stimulus. We used the autoreject algorithm [68] to interpolate bad channels, and we set the reference to the grand average. We then averaged, separately for each subject, all the available trials for each entity, in order to improve the signal-to-noise ratio in the evoked responses [62]. This left us with one evoked response per entity, to be used for the analyses.

The final preprocessing step was removing all the variance associated with word length from the EEG signal using cross-validated confound regression, which was validated in [69, 70].

## 2.4 Models

**2.4.1 Language models.** In order to create semantic representations for famous and personally familiar entities, we employed the texts described in Section 2.1 as inputs for two types

of language models, one static (word2vec) and one contextualized (XLM-Roberta-large). The rationale for this procedure is that words appearing in a text revolving around an entity carry a rich bundle of semantic information with respect to that entity. In other words, we interpret the texts we collected as entity-specific distributional lexical information: for personally familiar people and places, the answers to the questionnaires; for their famous counterparts, the text from their Wikipedia pages [41, 55, 56].

The procedure was matched across models, so as to avoid methodological confounds. For each individual entity, we split the text in passages of at least 20 words. We set a lower passage length threshold because, when encoding sentences for downstream use, LLMs have been shown to work better with rather long passages [71]. In order to ensure that text portions were long enough to work well with XLM, we rearranged the text so that sequential passages of at least 20 words were created (i.e. if a sentence were shorter than 20 words, we considered to be part of the same passage as the following sentence). We chose 20 words as a threshold since in English sentence lengths are most commonly is between 10 and 30 words [72].

After having encoded all passages of text using the language models, we retained only the vectors in the sequences corresponding to content words (i.e. open-class words: nouns, verbs, adjectives, adverbs) from the corresponding descriptive text (Wikipedia or questionnaire). We decided to follow [44] and exclude closed-class words, such as function words, since they do not carry semantic information. We reasoned that their presence would lead to the static language model being disadvantaged, since it has been shown that they struggle at representing function words properly [43, 73, 74].

Finally, we obtained a single entity representation in two steps. First, we averaged all the word vectors retained for each passage, thus obtaining one vector per passage. Then, we averaged all of the vectors for the individual passages of texts [32, 55, 75].

Therefore, at the end of the procedure we had one vector representation for each individual entity per model, capturing the distributional lexical information contained in our small-scale textual data.

We will now briefly describe the language models used. Word2vec is a feedforward neural network which learns vector representations (word vectors) from large-scale corpora [43]. Therefore, one single vector representation is created for each word type, regardless of homonymy and polysemy phenomena (e.g. the word 'bat' is modelled as a single vector, collapsing the two senses of animal and baseball instrument in a single representation). Such vector representations have been interpreted as models of cognitive semantic representations [11, 23, 76] and have been shown to capture well lexical processing [15, 16, 55]. We pre-trained a word2-vec model on the Italian version of Wikipedia (whose size was approximately 12 gigabytes). We used parameters suggested in the literature [15, 77]: skip-gram training regime, a vocabulary of 250000 words, a window size of 10 words and a vector size of 300 dimensions.

As a second model we used a contextualized language model [75], also often called a large language model (LLM). A contextualized language model is a deep neural network based on the Transformer architecture [46]. Its main difference with respect to a static language model is that, by design, it is aimed at the representation of words in context—i.e. sentences, paragraphs or longer passages. In LLMs, there are no 'static' word representations: in contrast, representations are adapted to each linguistic context used as input. This allows to capture fine-grained shifts in meaning, at both lexical and supra-lexical (e.g. discourse, dialogue) levels. Notice, however, that this is achieved at the cost of an extremely more sophisticated neural network architecture and costly training procedure [78]. Since publicly available LLMs for languages other than English are not on a par in terms of quality with their English counterpart [79], and our questionnaires were collected in Italian, we used a state-of-the-art multi-language model, XLM-Roberta-large [45]. Multi-language models are not specialized for

individual languages, however they can exploit transfer of cross-lingual semantic information during pre-training, a feature that can make them surprisingly effective [80, 81]. We chose XLM-Roberta-large because it is a widely used multilingual model with excellent performance both on cognitive [82] and NLP [83] tasks. Importantly, Italian was part of the languages contained in the pre-training corpus, thus making it a solid choice to encode text in Italian. It has 24 layers, 560 million parameters, and its vector dimensionality is 1024. It was trained on 2.5 terabytes of text covering 100 languages, including Italian, from a filtered CommonCrawl corpus [84]. We used the pre-trained model provided by Huggingface [85].

Since LLMs are designed to take as input natural language sentences, we used XLM to encode full sentences—but we averaged only the vectors corresponding to content words (see Section 2.4.1). Following previous work, we modelled entity representations using so-called (sentence) representation pooling for the top four layers [41, 75]—which consists of averaging the top four layers for each of the chosen words.

**2.4.2 Non-semantic baseline models.**  We also chose to report time-resolved encoding scores for two baseline, non-semantic models: name length and orthographic distance. We did so in order to show that the evoked responses did not contain signal that could be explained by such low-level variables. In the first model, we represented each individual entity by the length of their name. In the second one, we leveraged the Levenshtein distance, a popular measure of orthographic distance representing the number of letter substitutions required to transform one string into another [86]. We thus represented the orthographic properties of each name in a single value, as the average Levenshtein distance between the entity's name and all other names in the set of stimuli.

## 2.5 Encoding

**2.5.1 Representational Similarity Analysis encoding.**  For encoding, we used the approach proposed and validated in [87], which is based on Representation Similarity Analysis (RSA). We illustrate visually the RSA encoding procedure in Fig 2. The main advantage afforded by this methodology is that it ensures excellent performance without the need of fitting a model—which would be a concern given the small size of our dataset. RSA encoding is conceptually based on standard RSA [88], which we will here summarize shortly. Given a set of stimuli and a model that represents them in any quantitative form (numbers, vectors, etc.), the similarity between the brain responses to a given stimulus and its model representation can be measured in two steps. The first step is looking at how similar they are to all other representations in their own representational space (in the brain and in the model, respectively; the so-called first-order similarity). The second step consists of measuring directly the similarity between the two vectors of the pairwise similarities through the so-called second-order similarity. This approach has the advantage of providing a straightforward way to compare the representational structure of brains and models (*via* second-order similarity), something that would be otherwise difficult, given their difference in dimensionality.

In RSA encoding, as in most multivariate (encoding/decoding) approaches, the data is split iteratively in a train set and a test set [62, 89]. For each train/test split, the model predicts evoked responses for the test stimuli, which are compared with the real responses. A similarity metric (i.e. a correlation or distance measure) is used to evaluate how well the model captures brain activity. This procedure is repeated for all train/test split and correlations are then averaged, so as to provide an unbiased summary measurement [90].

As a similarity metric we use Spearman correlation between predicted and real evoked responses. We use Spearman correlation because it is nonparametric, thus making minimal assumptions about the relation holding between the model predictions and brain data; for this

# Representational Similarity Analysis Encoding

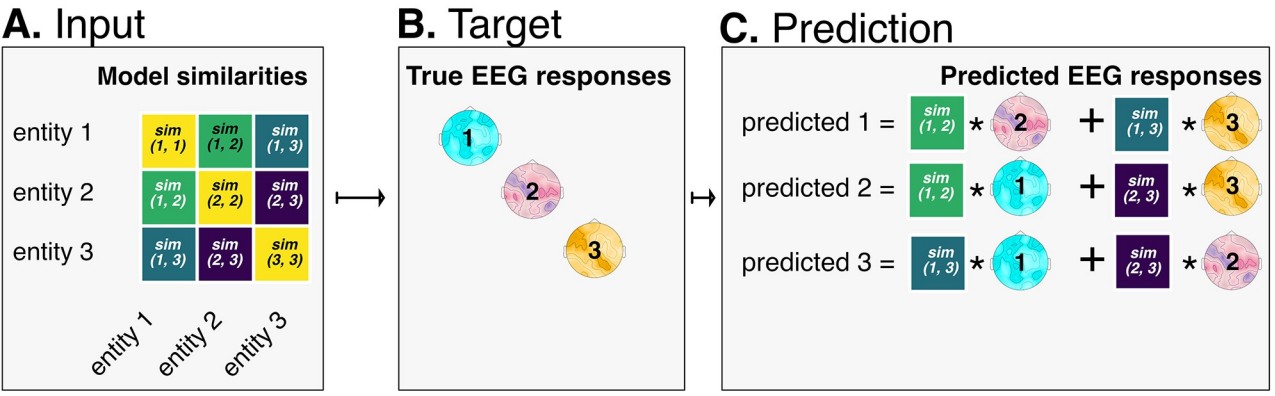

**Fig 2. Visualization of the RSA encoding procedure.** We visualize a toy example, where the whole dataset is composed of three exemplars. The input (part A) are similarities (e.g. Pearson correlations) in the model between entity representations. The targets (part B) are the corresponding EEG reponses for the same entities. Prediction happens in part C. For each entity (the test set, e.g. entity 1) its predicted EEG response corresponds to the the sum of the real evoked responses for the other entities (the train set, e.g. entity 2 and entity 3), where each response from the train set is weighted by its similarity in the model with the entity from the test set. Encoding performance is evaluated by computing the correlation between the predicted and the real response.

reason it is the recommended metric for comparing model predictions and brain data (i.e. second-order similarity) with RSA [88, 91].

Prediction is carried out in the following way. The evoked response to a given item from the train set is predicted as a weighted sum of the evoked responses to the stimuli in the training set. Following the original implementation, the weights to be used for the sum are the pairwise Pearson correlations between the test item and all of the training items in the model—in our case, a language model. Take for instance a toy training set composed of three individuals, for which both model representations $\vec{a} = Ana$, $\vec{b} = Milo$, $\vec{c} = Pati$, and their corresponding evoked responses $a_{brain}$, $b_{brain}$, $c_{brain}$ are available. Given a test item $d = Nico$, its evoked response $d_{brain}$ is predicted to be $d_{brain} = a_{brain}*r_{\vec{a},\vec{d}} + b_{brain}*r_{\vec{b},\vec{d}} + c_{brain}*r_{\vec{c},\vec{d}}$, where $r$ is the operation of Pearson correlation.

**2.5.2 Evaluation.** For the evaluation we followed standard practice in encoding/decoding brain studies [62, 90]. We carried out analyses separately for each individual subject (i.e. within subjects), and we then averaged results across subjects. Since for each subject we had limited data, we used a leave-two-out training/testing regime. This entails running iterated training/testing procedures, using each time as a test set one of all possible pairs of stimuli. At each iteration we recorded the Spearman correlation between each predicted test item and its true counterpart. The final correlation score was given by the average of all correlations collected by the leave-two-out train/test iterations.

**2.5.3 Confound control.** We controlled for name length for personally familiar people and places, since they could not be controlled *a priori* during stimulus selection, as was instead the case for famous names. To this aim, we used a cross-validated confound regression procedure that was previously validated [70] and used with this dataset in a category decoding setting [60]. For each train-test split, we fitted a linear regression model from the confound variable to the brain data within the train set. Then, only the residuals of the regression (for both the training and the test set) entered the analyses—effectively removing from the brain data the variance that could be explained by the confound variable. We use the original python implementation provided by the authors (https://skbold.readthedocs.io/en/latest/).

**2.5.4 Time-resolved encoding.** In our time-resolved analyses, we ran the encoding analysis separately for every time point [62] using all of the electrodes as target for the prediction. This whole-scalp approach allows to look primarily at how distributed patterns of evoked activity develop across time. By measuring the correlation between the real brain signal and the one predicted by the model, it is possible to understand when a model captures information as it is processed in the brain. Time points where correlations are above chance with statistical significance (using the TFCE method described in Section 2.5.6) indicate reliable encoding of such information.

**2.5.5 Spatio-temporal searchlight.** We were also interested in going beyond patterns of activity widely distributed across the whole scalp, and look at specific areas on the scalp where a model could explain evoked responses. To this aim, we implemented a searchlight encoding analysis. Searchlight allows to find in a bottom-up fashion localized clusters of brain activity associated with an experimental condition, while exploiting the high sensitivity of multivariate analyses [92, 93]. In practice, searchlight consists of running the encoding analyses repeatedly across smaller clusters of electrodes on the scalp.

To reduce the computational effort, we followed previous work [94] and used spatio-temporal clusters, where multiple time points are considered for each electrode within the cluster. As in [95], we employed a temporal radius of 50ms and a spatial radius of 30mm (i.e. a cluster contains evoked activity for 100ms, for electrodes falling within a circle having a diameter of 60mm). We computed statistical significance tests using the TFCE method described in Section 2.5.6. If the p-value for a cluster of electrodes at a given point in time fell below 0.05, we considered encoding to be significantly above chance.

**2.5.6 Statistical testing and corrections for multiple comparisons.** We ran one-tailed permutation statistical tests with the Threshold-Free Cluster Enhancement (TFCE) procedure, as implemented in MNE. This approach has two advantages. First of all, it is non-parametric, thus making minimal assumptions about the underlying data distribution. Secondly, it also controls by design for multiple comparisons, thus countering the risk of false positive due to the high number of statistical tests [62, 96, 97]. We tested the hypothesis that correlations between real and predicted scores were reliably above chance ($correlation_{chance}$ = 0.) across subjects, separately for each time point (time-resolved analysis) or spatio-temporal cluster (searchlight) within a time window between 0 and 800ms [64]. We set the significance threshold at 0.05, following conventional practice in the field [98].

# 3 Results

## 3.1 Comparing language models

In Fig 3 we report the overall encoding results for the time-resolved and searchlight analyses, respectively using all electrodes and localized clusters of electrodes. Encoding performance was measured as Spearman correlation among the predicted and the real evoked responses; the threshold used for statistical significance was $p < 0.05$ after TFCE correction for multiple comparisons. We also report the results of a direct comparison between XLM and word2vec, with the aim of finding out if, where and when the LLM can significantly outperform the static language model. In the following we will focus on data points reaching statistical significance; $T$ and $p$ values for all models and data points can be found together with the code and the data.

In the time-resolved analysis, for XLM, the contextualized language model, correlation scores were significantly above zero in a wide time window, from 250 to 800 ms (peak at 442.5ms, $T = 6.318$, $p < 0.001$). For the static language model, by contrast, the time window where the predicted responses correlated significantly with the real evoked potentials was

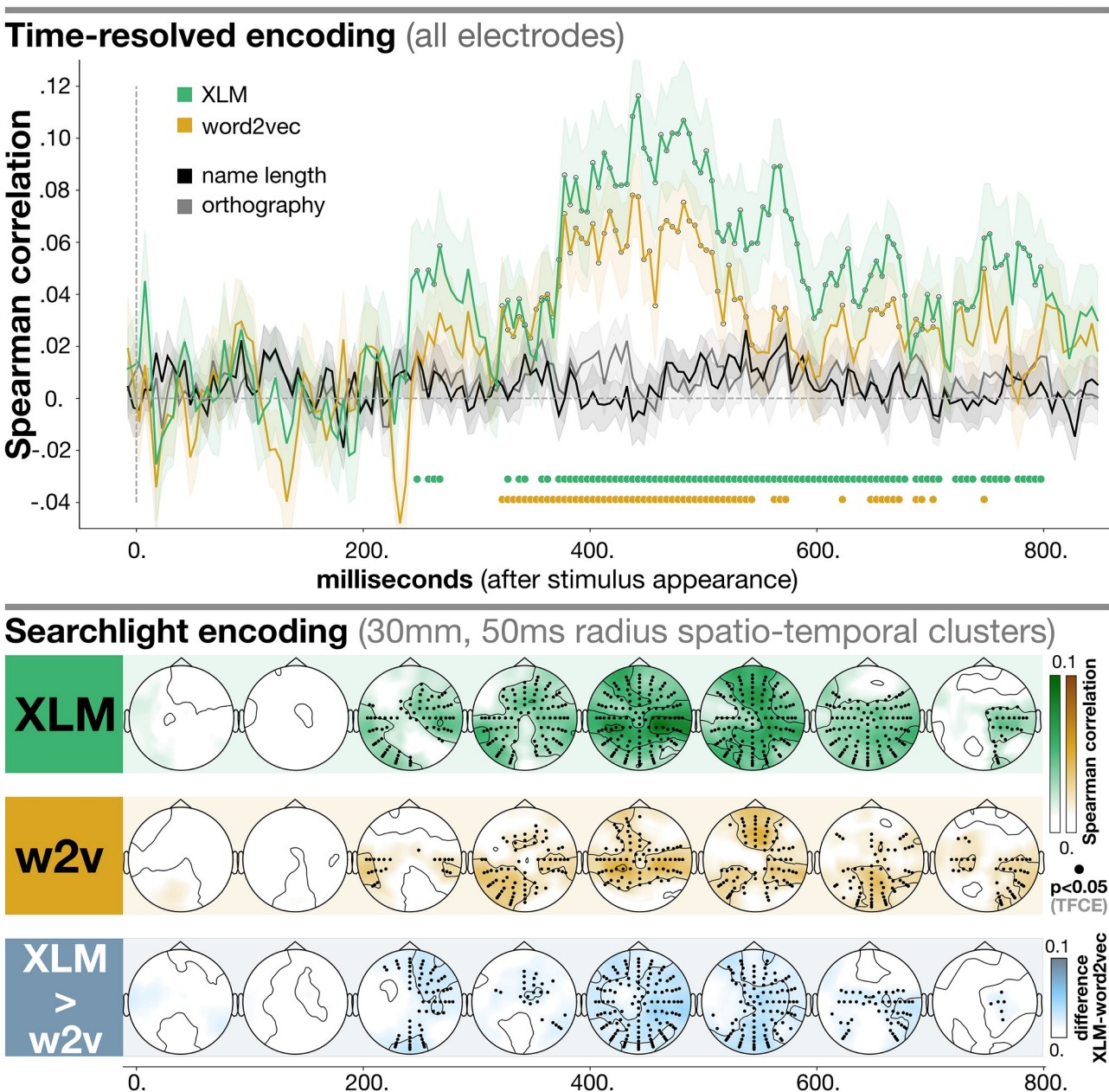

**Fig 3. Overall encoding performance for XLM and word2vec (w2v).** In the upper part of the plot we report time-resolved results obtained using all electrodes. The lower part contains searchlight results, which reflect encoding performance for smaller spatio-temporal clusters. In the final row we also report the results of the direct comparison between XLM and word2vec (i.e. the difference between XLM and word2vec). The evaluation metric used was Spearman correlation (on the y-axis for time-resolved analysis, color-coded for searchlight). Time (time-resolved) and space-time (searchlight) points where encoding (or, for the *XLM > word2vec* comparison, the difference among the two models) is significantly above chance ($p < 0.05$ after TFCE correction) are marked by thicker dots either at the bottom of the plot (time-resolved) or on scalp locations (searchlight). Results reported here are averaged across all entity types.

shorter, between 300 and 700ms (peak at 482.5ms, $T = 5.047$, $p < 0.001$). Also, across all time points, correlation was consistently higher for XLM when compared to word2vec.

Regarding the two baseline non-semantic models (name length and orthography), correlations in the time-resolved analyses were never significantly above chance (peak for name length: 572.5ms, $T = 1.449$, $p = 0.145$; peak for orthography: 432.5ms, $T = 1.086$, $p = 0.403$). This confirms that such confounds were successfully removed from the signal through the confound removal procedure.

In searchlight, the predictions from XLM showed significant correlations from 200 to 800ms. Clusters could initially (200–300ms) be found in both hemispheres, in temporo-parietal electrodes (left peak: D22, $T = 14.307$, $p < 0.001$; right peak: B17, $T = 14.005$, $p < 0.001$). Encoding with XLM kept providing significant scores in bilateral electrodes up until 700ms (left peak: D20 between 500–600ms, $T = 13.106$, $p < 0.001$; right peak: B11 between 500–600ms, $T = 15.846$, $p < 0.001$). After that time point, significant clusters were right lateralized, again in temporo-parietal regions (peak: B17 between 700–800ms, $T = 12.403$, $p < 0.001$). Correlations in central regions were significant in smaller time windows—frontally, between 300 and 600ms (peak: C28 between 500–600ms, $T = 17.452$, $p < 0.001$); posteriorly, between 400 and 700ms (peak: A30 between 600–700ms, $T = 16.162$, $p < 0.001$).

For searchlight encoding with word2vec, clusters where correlation was significantly above zero emerged from 200–300ms (overall peak: C21 between 500–600ms, $T = 11.544$, $p < 0.001$). Significant clusters for the static language model followed a rather similar spatial and temporal distribution with respect to XLM: bilateral temporo-parietal electrodes between 200 and 800ms (left peak: D24 between 300–400ms, $T = 10.745$, $p = < 0.001$; right peak: B16 between 400–500ms, $T = 11.241$, $p < 0.001$); fronto-central electrodes between 300 and 600ms (peak: C21 between 500–600ms, $T = 10.544$, $p < 0.001$); posterior-central electrodes between 400 and 700ms (peak: A26 between 700–800ms, $T = 10.003$, $p = 0.002$).

Aside from such general trends of similarity, significant differences were revealed among the two models by the direct comparisons reported in the final row of Fig 3. XLM performed significantly better in a very large amount of clusters between 400–600 (peak difference: B18 between 400–500ms, $T = 12.655$, $p < 0.001$). Also, the encoding scores of XLM were significantly higher in central frontal and posterior electrodes between 200–300ms (frontal peak: C14, $T = 10.157$, $p = 0.002$; posterior peak: A23, $T = 8.405$, $p = 0.0146$). However, between 300–400ms and 700–800ms the difference showed a much more restrained spatial distribution. This converges with the time-resolved results (see upper portion of Fig 3), where it can be seen that the difference in performance in those time ranges was smaller than in other time points. Between 300–400ms, significant differences were mostly localized around the vertex and the right hemisphere (peak: C1, $T = 9.464$, $p = 0.002$). Between 700–800ms, by contrast, the largest difference was found in a right temporo-parietal cluster (B20, $T = 9.424$, $p = 0.003$).

## 3.2 Encoding scores for specific types of entities

In Fig 4 we detail how the best-performing model, XLM, captures semantic information for each specific type of individual entity (personally familiar places, famous places, personally familiar people, famous people). In Fig 5 we report the equivalent set of results for word2vec—thus allowing to better understand what categories drive the differences in performance among the models. The goal of this analysis was understanding whether our small-scale distributional information, as encoded by language models, worked equally well for all types of entities, despite their obvious differences.

To report scores separately for each type of entity, we repeated the encoding analyses using as test items only the stimuli belonging to the relevant category. The rationale is that

# XLM - Detailed encoding evaluation

## Time-resolved encoding (all electrodes)

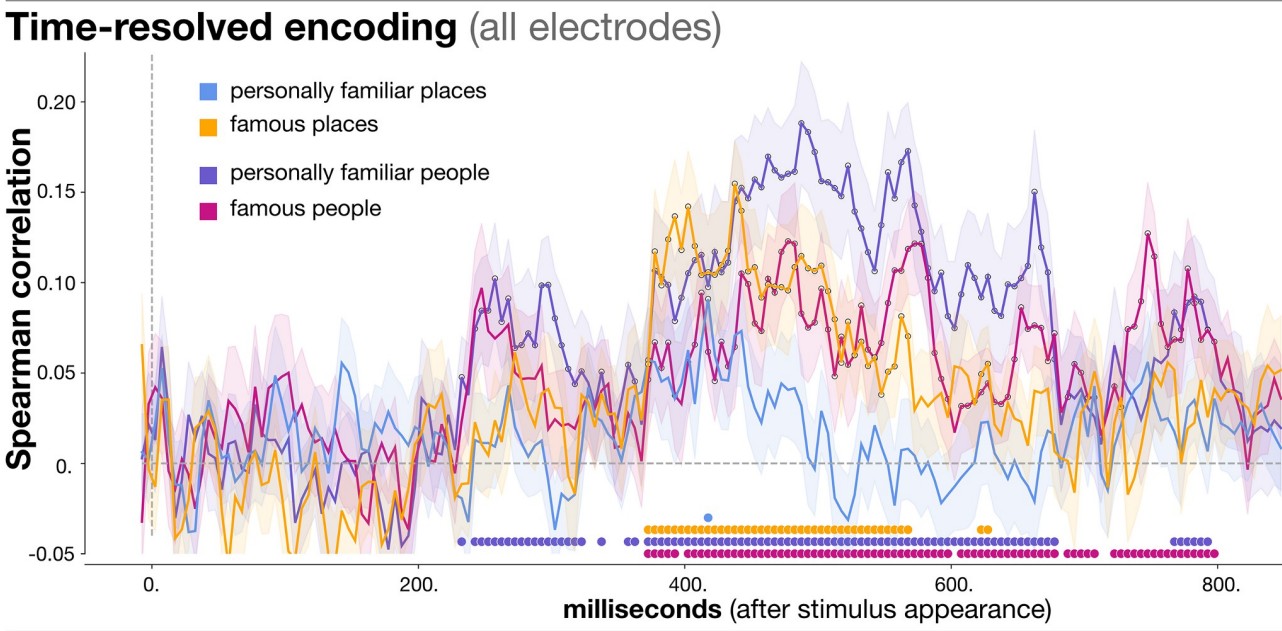

## Searchlight encoding (30mm, 50ms radius spatio-temporal clusters)

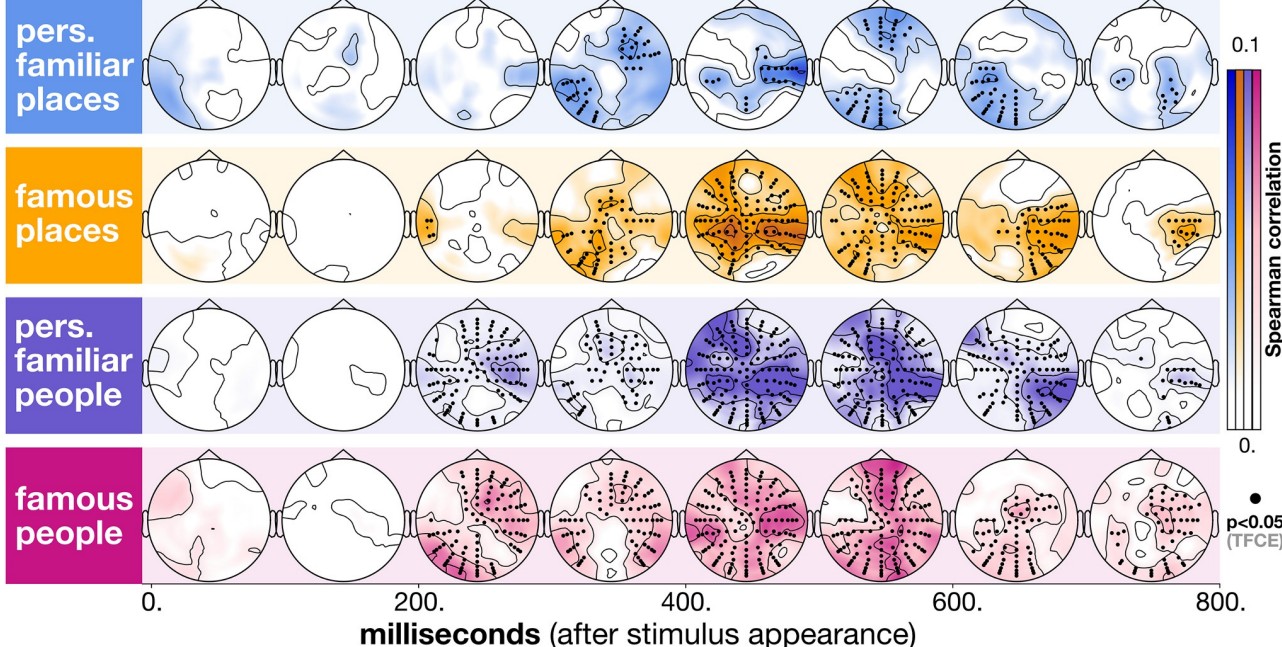

**Fig 4. Detailed evaluation of encoding performance for XLM for all categories and levels of familiarity.** In the upper part of the plot we report time-resolved results obtained using all electrodes. The lower part contains searchlight results, which reflect encoding performance for smaller spatio-temporal clusters. The evaluation metric used was Spearman correlation (on the y-axis for time-resolved analysis, color-coded for searchlight). Time (time-resolved) and space-time (searchlight) points where encoding is significantly above chance ($p < 0.05$ after TFCE correction) are marked by thicker dots either at the bottom of the plot (time-resolved) or on scalp locations (searchlight). Results reported here refer to XLM only, and are separated for each entity type.

# word2vec - Detailed encoding evaluation

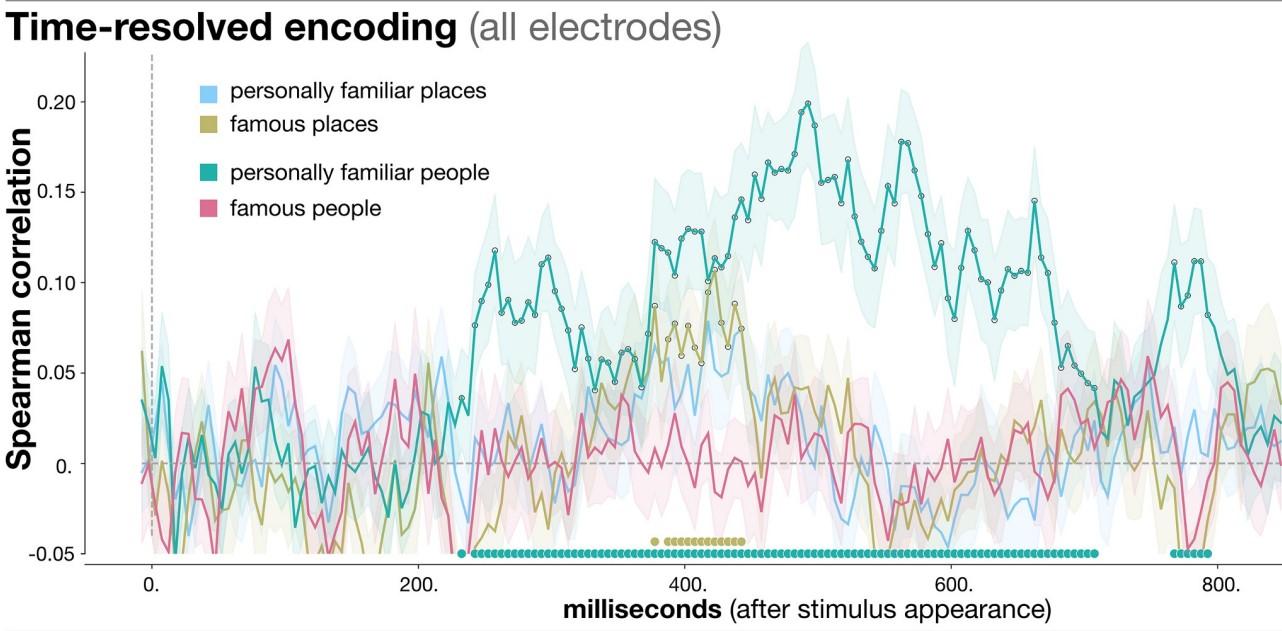

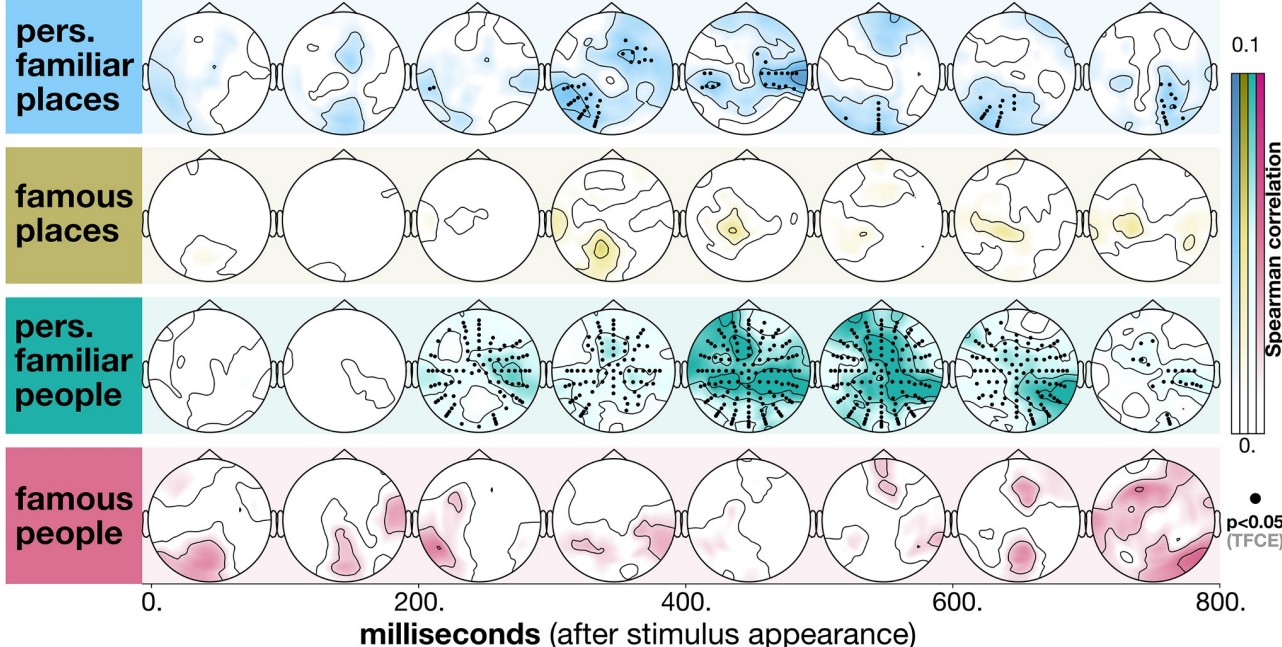

**Fig 5. Detailed evaluation of encoding performance for word2vec for all categories and levels of familiarity.** In the upper part of the plot we report time-resolved results obtained using all electrodes. The lower part contains searchlight results, which reflect encoding performance for smaller spatio-temporal clusters. The evaluation metric used was Spearman correlation (on the y-axis for time-resolved analysis, color-coded for searchlight). Time (time-resolved) and space-time (searchlight) points where encoding is significantly above chance ($p < 0.05$ after TFCE correction) are marked by thicker dots either at the bottom of the plot (time-resolved) or on scalp locations (searchlight). Results reported here refer to word2vec only, and are separated for each entity type.

performance on a test item is a quantification of the extent with which a model is able to capture the specific phenomenon of which the test item is an instance [31, 47, 99–103].

Both time-resolved and searchlight results indicate that personally familiar people were the type of entities best captured by XLM, and personally familiar places the worst. For personally familiar people, correlation was significantly above zero between 200 and 800ms in a large bilateral set of electrodes, concentrated in temporo-parietal and fronto-central areas (temporo-parietal peaks: for the right hemisphere, B17 between 400–500ms, $T = 16.511$, $p < 0.001$; for the left hemisphere, A8 between 500–600ms, $T = 16.061$, $p < 0.001$; fronto-central peak: C26 between 500–600ms, $T = 16.612$, $p < 0.001$). Interestingly, right-hemisphere temporo-parietal electrodes provided the highest and most consistent encoding scores overall (as reported above, around B17). For personally familiar places the only time point in the time-resolved analysis where statistical significance was reached was 417.5ms ($T = 1.982$, $p = 0.04$). However, searchlight revealed several clusters where encoding was reliably above chance. Such clusters were first aligned over a bilateral temporo-parietal axis (left peak: A8 between 500–600ms, $T = 7.711$, $p = 0.0263$; right peak: B25 between 400–500ms, $T = 7.54$, $p = 0.03$), and later over a central frontal-to-posterior axis (frontal peak: C17 between 500–600ms, $T = 7.26$, $p = 0.047$; posterior peak: A24 between 500–600ms, $T = 7.572$, $p < 0.029$).

Famous entities showed a less dramatic difference across categories: in the time-resolved setting, evoked responses for famous people could be encoded significantly better than chance between 350 and 800ms (peak at 557.5ms, $T = 4.384$, $p < 0.001$), whereas famous places did so in the 350–650ms range (peak at 437.5ms, $T = 4.482$, $p < 0.001$).

Searchlight encoding for XLM revealed both differences (before 400ms and after 500ms) and similarities (between 300–400ms) in spatial patterns across famous people and places. For famous places, clusters where encoding was significantly better than chance appeared mostly in temporo-parietal bilateral areas (left peak: D22 between 400–500ms, $T = 9.307$, $p = 0.005$; right peak: B17 between 400–500ms, $T = 10.474$, $p = 0.002$). The involvement of frontal electrodes was minor, between 400 and 600ms (peak around C28 between 500–600ms, $T = 9.486$, $p = 0.004$). Finally, after 600ms, a clear right lateralization was found (peak around B16 between 600–700ms, $T = 9.737$, $p = 0.001$). For famous people, if until 500ms encoding performance with XLM was statistically significant in bilateral temporo-parietal clusters (left peak: D26 between 400–500ms, $T = 13.423$, $p < 0.001$; right peak: B24 between 400–500ms, $T = 14.544$, $p < 0.001$), later it expanded to central frontal and posterior electrodes (frontal peak: C14 between 500–600ms, $T = 14.073$, $p < 0.001$; posterior peak: A15 between 500–600ms, $T = 13.355$, $p < 0.001$), similarly to what happened for personally familiar people.

The results obtained for word2vec with the same approach (Fig 5) were quite different, revealing which types of entities determined its overall lower performance (see Fig 3). In the time-resolved setting, encoding performance was above chance only for personally familiar people (peak at 467.5ms, $T = 7.014$, $p < 0.001$) and famous places (peak at 417.5ms, $T = 2.405$, $p = 0.007$), and approached significance for personally familiar places (peak at 437.5ms, $T = 1.663$, $p = 0.095$). By contrast, word2vec did not seem able to explain evoked responses for famous people (peak: 747ms, $T = 0.908$, $p = 0.568$). When looking at individual clusters using searchlight, patterns were similar to XLM for personally familiar entities, but different for famous entities. In particular, for the former, the encoding performance of word2vec was significant in a set of clusters largely correspoding to those emerging from the matched analyses with XLM. In the case of personally familiar people, this consisted of temporo-parietal and frontal electrodes (left peak: D23 between 400–500ms, $T = 15.568$, $p < 0.001$; right peak: B17 between 400–500ms, $T = 16.521$, $p < 0.001$; frontal peak: C20 between 500–600ms, $T = 16.655$, $p < 0.001$); for personally familiar places, bilateral temporo-parietal clusters first (left peak: A8 between 300–400ms, $T = 8.032$, $p = 0.015$; right peak: B26 between 400–500ms, $T = 7.698$,

$p = 0.027$), and centro-posterior later (peak: A14 between 600–700ms, $T = 7.413$, $p = 0.033$). For famous entities, performance was never above chance, for any cluster (peak for famous people: B7 between 700–800ms, $T = 4.138$, $p = 0.541$; peak for famous places: A5 between 300–400ms, $T = 3.01$, $p = 0.801$).

This indicates that word2vec interacted differently, depending on the category, with distributed and localized patterns of evoked activity: while in the case of famous places distributed activity was crucial in reaching significance (i.e. encoding was significant only in the time-resolved, but not in the searchlight analysis), for personally familiar places the opposite was true.

## 4 Discussion

### 4.1 Personal memories as a window into semantic processing of individual entities

The main contribution of this work is showing that personal memories revolving around a subject's closest people and places, collected from participants through a questionnaire, can be used to create semantic representations with language models that capture how the brain represents those very same individuals.

Our approach was motivated by the hypothesis that the way in which we talk about the world reflects our internal cognitive states [104]—and, more specifically for semantics, the idiosyncratic perspective that shapes the way in which we see the world and represent concepts [37]. Such hypothesis is supported by previous results showing that textual analyses of a speaker's utterances can be used to uncover a speaker's unique perspective on concepts [32, 105], their emotions [106, 107], their personality traits and mental health [108–113].

Time-wise, encoding scores were statistically significant in the 200–800ms range, with peaks at around 400ms and 700ms, which is where traditionally semantic and memory retrieval processing has been found in EEG studies—specifically, N400 [114] and late posterior complex, LPC [115].

Spatial patterns of encoding emerging from the searchlight analysis highlight the role, overall, of temporo-parietal bilateral regions of the scalp in the 400–500ms range. Such electrode clusters are situated roughly above portions of the cortex which have been shown to be crucial for the processing of individual entities by a vast literature from neuroimaging (summarized in [116]). This provides new evidence that markers of processing of proper names on the scalp seem to follow a similar trajectory as cortical activity (Fig 3, lower portion; see also [64] for similar decoding maps).

This consistent pattern of results therefore confirms our hypothesis. The distributional lexical information contained in the questionnaire answers, describing the three main components of memories for individual entities—episodic and semantic knowledge, as well as personal semantics—can be used to encode how the brain represents those individual entities.

### 4.2 Using language models to encode personally familiar entities

We were able to encode responses to individual entities by exploiting, through language models, the distributional lexical information contained in the questionnaire answers. Our results thus indicate that the semantic representations obtained from language models are rich, multi-faceted models of brain processing (cfr. [11, 117]) and that, following an appropriate methodology, they can be combined to create vectors for entities that do not appear in their pre-training data [27, 32]. The questionnaire was composed of nine questions only, to which

participants had to answer to with open-ended natural sentences—which by NLP standards is extremely small [24].

This result is all the more surprising given not just the small size of the textual data, but also the fact that participants produced answers freely, without having to stick to categories or dimensions defined a priori by the experimenter as was the case in previous neuroimaging studies [33–35].

A key role in making the most of such reduced textual data was played by a dedicated vector creation methodology (see Section 2.4.1). We created original semantic representations for personally familiar individual entities from the unconstrained questionnaire answers, building upon the previously-acquired distributional lexical knowledge contained in language models. In other words, through language models we could model semantic knowledge of people and places as idiosyncratic combinations of pieces of generic knowledge (the pre-trained word vectors), shaped and guided by autobiographic experiences (the person-specific words contained in the answers to the questionnaires).

In this respect, XLM, the contextualized language model, showed consistently better performance, when compared to word2vec, a static language model, at capturing semantic processing in the brain (Fig 3). Notice that we matched the vector extraction methodology (Section 2.4.1), thus levelling as much as possible the differences between the two models. Two remaining factors differentiate static and contextual models. The first one is related to their size, both in terms of the amount of distributional information used during training (2.5 terabytes for XLM, 12 gigabytes for word2vec), and of their number of parameters (560 million parameters for XLM, 75 millions for word2vec—computed as in [118] as the multiplication of the number of vector dimensions by the vocabulary size; see Section 2.4.1). The second one is that contextualized models are able to adapt their vector representations to specific linguistic contexts, capturing fine-grained contextual shifts in meaning, whereas static models cannot [75]. These two features, therefore, seem to be important in order to capture the semantic information revolving around an individual entity, which is an extremely idiosyncratic and unique mix of features and memories [1, 2]. Notice that, looking the current literature, it is debated which of these factors—and why—could play a bigger role when it comes to predicting cognitive processes (for evidence in opposite directions, see [119–122]); more work will be required to disentangle the two. Nevertheless, this evidence dovetails with previous results in the literature indicating that LLMs can improve over static models when predicting generic concepts and famous individual entities [31, 48, 123, 124].

Finally, it should be underlined that, despite its overall inferiority, a static model like word2vec could nevertheless reach statistically significant encoding performance, both overall and for some individual types of entities (Figs 3 and 5). This can provide some reassurance over the ability of simpler, lighter static models to efficiently deal with distributional semantics information [125], even as fine-grained as personal memories—an important point for future studies involving low-resource settings and languages [126].

## 4.3 Individual entities

Detailed patterns of encoding performance can be compared in terms of scores and spatial location (Fig 4) across semantic categories (people and places) and levels of familiarity (famous and personally familiar). This allows to obtain insights with respect to the way in which the brain represents individual entities.

First of all, it is important to notice that, as experimental stimuli, we used names, instead of images—which is by contrast the most common choice when looking at concepts in the brain [63], possibly because they afford higher encoding/decoding performance [18, 64, 127, 128].

The reason why we chose to use names in our experiment is that this would allow us to elicit semantic processing of people and places in the brain not biased towards any sensory modality. For instance, had we used images as stimuli, we would have had two types of non-semantic, visual confounds. First, images clearly differ in terms of visual properties across people and places and therefore evoke strongly distinct responses [129, 130]. Secondly, using a picture elicits brain activity associated with that specific instance of the picture and its low-level visual features [131, 132].

Overall, we found that the semantics of people, both personally familiar and famous, could be captured by XLM, a large language model, using distributional lexical information representing declarative memories (in its three components—semantic, episodic and personal)—despite the uniqueness of the semantic information for each individual [1–3, 116]. The performance of word2vec, a static language model, was worse than XLM. In particular, it seemed to interact more strongly with the type of entity to be represented, being consistently lower for famous entities.

Turning to places, a more mixed set of results emerges. For XLM, a particularly striking difference in terms of encoding performance was found between personally familiar people and places (Fig 4): performance was barely above chance for places, while it was significant in a large time window for people. Interestingly, the patterns of encoding were quite similar for personally familiar people and places also for word2vec (Fig 5).

At least two factors may be hypothesized to be at work. First of all, it is possible that the questionnaire could not capture place-specific types of information—for instance, sensory and modality-specific features—that are crucial when it comes to cognitive processing of familiar places. Secondly, as discussed in [2, 60], the identity of personally familiar places may be harder to process than that of personally familiar people—for various reasons that could be evolutionary [133], social [134], or related to the availability of semantic features during retrieval [135]. This would then make it harder to correctly distinguish place-specific, as opposed to person-specific, signatures in brain activity—a result that converges with the overall lower encoding scores for places reported here (Fig 5) and that has previously been found in the literature [60, 135, 136]. While we are unable to provide an answer to such questions with the current experiment, we believe this could be a fruitful direction for future research.

Furthermore, searchlight encoding, which provides a window on focal activity on the scalp, highlights both commonalities and differences for the various types of individual entities.

When looking at the commonalities, a core set of spatio-temporal clusters shared across models, categories and levels of familiarity emerges, indicating general correlates of semantic processing of individual entities. When using XLM, the best-performing language model, encoding was significant for all type of individual entities in the range between 300 and 700ms, first in temporo-parietal bilateral electrodes (300–500ms), then fronto- and posterior- central electrodes (500–700ms). Similar patterns were found for word2vec when aggregating across all types of entities. Temporo-parietal bilateral clusters are typically associated with general semantic processing in the N400 time range [63, 114].

By contrast, the importance of frontal and central electrodes could be explained as an involvement of both Default Mode Network areas, which are activated by episodic memory and social processing [137–140], and of posterior visual areas relevant for processing mental imagery [141, 142].

Turning to the differences, the earliest time range where correlations were significant (200–300ms) revealed an interesting pattern: a stronger presence of information related to people as opposed to places in the fronto-temporal right hemisphere (Fig 4, lower portion). This can be connected to results coming from the neuroimaging literature, where it has been shown that the right hemisphere is crucial specifically for person identity processing [135, 143–146]. Also,

this seems to converge with previous results showing person-selective early evoked activity [60], suggesting that not only quantitative, but also qualitative differences exist among the two (i.e. differences not only in terms of encoding performance, but also of neural processes involved; cf. above).

On a more abstract, philosophical level, our findings would seem to be compatible with so-called **descriptivism**. Descriptivism is a view on proper names and individual entities that proposes that the meaning of a proper name can be in fact equated to a set of sentences describing that entity (in our case, declarative memories) [147, 148]. After all, we have shown that the words contained in short texts about individual entities can be used to capture relevant aspects of the way in which brains process their representations. However, one could also argue, as pointed out by [116], that descriptions (and by extension, declarative knowledge) fail at capturing all semantic information associated with cognitive representations of individual entities, namely not considering socio-emotional dimensions.

We believe that it is precisely our choice of modelling memories (i.e. descriptions) through language models that allows to address this concern. Language models have been consistently shown to capture both social and emotional dimensions of word meaning from lexical distributional information [11, 13, 14]. Therefore, our view is that although social and emotional semantic dimensions of individual entities can be studied in isolation (cfr. [33–35]), they are latent in declarative memories (descriptions) and can be therefore be captured in the mixture of semantic information and dimensions present in language models [117]. In this sense, descriptions are a richer source of semantic information about individual entities than it may appear at first sight: they not only convey their propositional content, but also the broader semantic information hidden in the words that make them up.

## 5 Conclusion

In this work we have shown that it is possible to capture semantic knowledge about personally familiar individual entities by extracting semantic representations from subject-specific memories using language models. Importantly, we also demonstrated that similar performance could be obtained also for a matched set of famous entities, thus proving the solidity of our approach. The results of our multivariate encoding analyses indicate that entity-specific information emerges in a time window usually associated with semantic processing, between 200–800ms. Also, this seems to be the case especially in bilateral temporo-parietal regions, which converges with neuroimaging studies. Overall, our results exploit cutting-edge models in AI to provide a window into extremely fine-grained, subject-specific semantic knowledge as it is processed in the brain. We hope that this will motivate future work aiming at the investigation of individual uniqueness in semantic processing.

## Supporting information

**S1 File.**
(PDF)

## Acknowledgments

We would like to thank prof. Davide Crepaldi, head of the Language, Learning and Reading lab at SISSA, who provided the facilities for collecting the EEG data while the first author was visiting, and Marjina Bellida for helping out with subject preparation procedures. Finally, Natalia Ginzburg, from whom we borrowed, out of admiration for the wonderful book 'Lessico Famigliare', part of the title of this paper.

## Author Contributions

**Conceptualization:** Andrea Bruera, Massimo Poesio.

**Data curation:** Andrea Bruera.

**Formal analysis:** Andrea Bruera.

**Funding acquisition:** Massimo Poesio.

**Investigation:** Andrea Bruera.

**Methodology:** Andrea Bruera.

**Project administration:** Andrea Bruera, Massimo Poesio.

**Resources:** Massimo Poesio.

**Software:** Andrea Bruera.

**Supervision:** Andrea Bruera, Massimo Poesio.

**Validation:** Andrea Bruera.

**Visualization:** Andrea Bruera.

**Writing – original draft:** Andrea Bruera.

**Writing – review & editing:** Andrea Bruera, Massimo Poesio.

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
