## [Decision Letter · Decision Letter 0]

14 May 2024

PONE-D-23-26777Family Lexicon: using language models to encode memories of personally familiar and famous people and places in the brainPLOS ONE

Dear Dr. Bruera,

Thank you for submitting your manuscript to PLOS ONE. After careful consideration, we feel that it has merit but does not fully meet PLOS ONE’s publication criteria as it currently stands. Therefore, we invite you to submit a revised version of the manuscript that addresses the points raised during the review process.

We look forward to receiving your revised manuscript.

Kind regards,

Michal Ptaszynski, PhD

Academic Editor

PLOS ONE

 [AB was supported by a doctoral studentship from the School of Electronic Engineering and Computer Science, Queen Mary University of London.].  

5. We note that you have indicated that there are restrictions to data sharing for this study. PLOS only allows data to be available upon request if there are legal or ethical restrictions on sharing data publicly. For more information on unacceptable data access restrictions, please see http://journals.plos.org/plosone/s/data-availability#loc-unacceptable-data-access-restrictions. 

Reviewers' comments:

Reviewer's Responses to Questions

**Comments to the Author**

1. Is the manuscript technically sound, and do the data support the conclusions?

Reviewer #1: Yes

Reviewer #2: Yes

Reviewer #3: Yes

Reviewer #4: Yes

2. Has the statistical analysis been performed appropriately and rigorously? 

Reviewer #1: Yes

Reviewer #2: Yes

Reviewer #3: Yes

Reviewer #4: Yes

3. Have the authors made all data underlying the findings in their manuscript fully available?

Reviewer #1: No

Reviewer #2: Yes

Reviewer #3: No

Reviewer #4: Yes

4. Is the manuscript presented in an intelligible fashion and written in standard English?

Reviewer #1: Yes

Reviewer #2: Yes

Reviewer #3: Yes

Reviewer #4: Yes

5. Review Comments to the Author

Reviewer #1: The research domain and rationale are clearly defined, thanks to a rich and detailed Introduction section. The overall objective proposes a contextualised, large-language model-based approach to capture fine-grained personally-familiar entities (places and expecially persons) better than more tradiotonal static language models (such as those using word2vec representations).

Results support the suggested approach, in particular for what it concerns personally familiar people; whereas worse results are shown for personally familiar places.

In such account, the proposed explanation is not totally convincing. My suggestion is to clearly state that more analyses should be done to clarify to clarify the differences in encoding places versus people.

If the Introduction section is clearly written, the other sections need small revisions from yours:

- line 115: Fig 1 left portion  right portion

- line 116: We  we

- lines 116-117: .. had not either ... or  had neither .. nor

- line 128: pleade add standard deviations as well

- line132 and all ensuing occurrences: see Section  reference to Section is always missing

- line 248: (Wikipedia or questionnairre)  why Wikipedia? in lines 155-157 I read that for individual entity you take names and a short description (occupations for people and type of places)

- lines 251-254: here the syntactic structure is a bit odd and access to the content is quite difficult.

- line 274: at at (remove one of them)

- line 291: odd punctuation (sentence; and so on  sentence, and so on)

- lines 295-302: pay attention to the consecutio temporum.

Reviewer #2: Regrettably on reading the full paper I discovered that I am not competent to review it becaue there is virtually no linguistics in it but computational encephelography. I apologize for any inconveneince caused by my inept decision. Although I have recommended rejection it was a forced decision and should be ignored.

Reviewer #3: Thank you for letting me review this very interesting paper on how language models can encode subject-specific information. The topic itself is very interesting because usually language models try to capture concepts from large amounts of data, while in this case, we see how they can also be used to encode some fragments of personal life. The manuscript is well written and the study is properly evaluated.

* Is the manuscript technically sound, and do the data support the conclusions?

Yes.

* Has the statistical analysis been performed appropriately and rigorously?

The methodology used in the study is derived from the literature and appears soundly performed.

* Have the authors made all data underlying the findings in their manuscript fully available?

No, the data obtained from the study contain subject information that is not publicly available. However, the data used for the analysis have been made available.

* Is the manuscript presented in an intelligible fashion and written in standard English?

Yes, the manuscript is well-written. I have pointed out below a couple of sentences that could be improved for easier reading.

Specific comments:

Abstract:

- I suggest to rewrite the sentence "In terms of spatio-temporal clusters, ... (500-700ms)." for better clarity.

Methods -> Famous entities:

- Lines 127-139: Have you checked that the assumptions of the t-test are met?

Methods -> Personally familiar entities:

- In line 154, you mention that you provided the text of the specific instructions given to the subjects in the code. Scrolling quickly through the repository I did not find it. I suggest that you 1) Update the Readme so that it is easier to find the information and 2) Provide a list of the questions/instructions as part of the manuscript (in the appendix?). I think it is a very interesting part of the study and that it should be included.

Methods -> Spatio-temporal searchlight:

- I suggest to rewrite the sentence in lines 371-373: "Results are intepreted as above... at a given point in time" for better clarity.

Methods -> Statistical testing:

- Line 380: We ran one-tailed statistical tests: which ones?

Results:

- It would be interesting to see the same figure as Figure 4, but for word2vec, so that it would also be possible to compare how this model performed for the different tasks.

- Figure 3, Figure 4: it would be nice to add that the time radius for the searchlight was 50ms.

- Figure 3, Figure 4: it would also be nice to add the x-axis under the searchlight coding.

- Figure 4: What do the points indicated as stars in the graph represent?

Other comments:

- The hyperlinking of the sections did not work, see lines 132, 158, 178, 182, 240, 284, 497, 510,...

- Line 204: "Each trial proceeded as followS". Missing S.

- The acronym MNE is never spelled out.

- Line 285: "pooling for the top four layerS". Missing S.

- Figure 3 and Figure 4: space missing between "scalp locations" and "(searchlight)".

Reviewer #4: In this paper, the authors use language models(both small and big model) to build a connection between semantic knowledge from personally familiar individual entities and semantic information in brain. And a series experiment show that encoding performance was significant in a large time window (200-800ms).I think the research topic is very interesting and owns practical value. There are some questions as follows, (1) why you choose 10 words to to form a topic? (2) It seems some hot words could own high weights and strong connections because experiment performance is better by big language than small model, have you try to test key-word's effect in your model?

6. PLOS authors have the option to publish the peer review history of their article (what does this mean?). If published, this will include your full peer review and any attached files.

Reviewer #1: **Yes: **Claudia Marzi, ILC-CNR

Reviewer #2: No

Reviewer #3: No

Reviewer #4: No

---

## [Author Response · Author response to Decision Letter 0]

16 Aug 2024

To the attention of the Editorial Board of PLOS ONE,

We are now ready to submit a revised version of our paper. We’d like to thank the editor, for arranging the reviewers' responses to our manuscript, and the reviewers, whose suggestions have helped us significantly in improving the quality of the paper and in making it clearer.

We have taken into consideration all the comments, suggestions and concerns raised by the reviewers, and made the required revisions.

We have reported all the individual answers to the reviewers in the next pages. In the response to the reviewers, the original reviewers’ comments are indicated by double quotation marks; our answer is in bold italics; the relevant sections from the revised manuscript are reported in italics (after the words 'NEW TEXT' or 'ADDED TEXT').

Summarizing the main changes, 

First of all, we have added a full figure (Figure 5), reporting complete, in-depth results for word2vec. This allows to see the differences between this model and the large language model, as suggested by a reviewer; 

Secondly, we have addressed the concerns put forward by two of the reviewers about the obscurity of some parts of the methodological details - namely related to the creation of the vectors for the entities, and to the experimental procedure. The relevant sections have been largely re-written; 

Thirdly, we have substituted t-tests - whose assumptions, as correctly pointed out by a reviewer, may not have been valid with our data - with non-parametric permutation tests, and corrected all relevant sections. We have also substituted the measure of Pearson correlation, that presented similar issues, with that of Spearman correlation, when comparing the model's predictions and the real EEG data;

Then, we have explicitly acknowledged the limitations pointed out by one of the reviewers in the discussion;

We have added as Supplementary Information a pdf containing a translation to English of the original questionnaire;

Finally, we have added all details required by the reviewers to the figures, and we have changed all the colour palettes so as to make them readable also by people affected by colour blindness.

In the version of the manuscript showing the changes between the first submission and the current revision, you will find the deleted text striked through and in blue; the added text in dark orange.

 We hope that you will find our responses satisfactory.

 Yours sincerely,

Andrea Bruera & Massimo Poesio

Response to the reviewers - in the following, we respond to all the points raised by the reviewers, reporting the original text preceded by an acronym for the reviewer (e.g. 'R1: ' for reviewer 1), and our answer preceded by 'answer: '.

Reviewer #1 (R1): 

R1: "The research domain and rationale are clearly defined, thanks to a rich and detailed Introduction section. The overall objective proposes a contextualised, large-language model-based approach to capture fine-grained personally-familiar entities (places and especially persons) better than more traditional static language models (such as those using word2vec representations)."

Answer: Thanks for your time and for raising very relevant points that, we believe, allowed us to noticeably improve the quality of this work.

R1: "Results support the suggested approach, in particular for what it concerns personally familiar people; whereas worse results are shown for personally familiar places.

In such account, the proposed explanation is not totally convincing. My suggestion is to clearly state that more analyses should be done to clarify to clarify the differences in encoding places versus people."

Answer: Thanks for the suggestion. We have now added a paragraph where we discuss this point, and we acknowledged explicitly the current limitations.

ADDED TEXT:

"Turning to places, a more mixed set of results emerges. For XLM, a particularly striking difference in terms of encoding performance was found between personally familiar people and places (Fig 4): performance was barely above chance for places, while it was significant in a large time window for people. Interestingly, the patterns of encoding were quite similar for personally familiar people and places also for word2vec (Fig 5).

At least two factors may be hypothesized to be at work. First of all, it’s possible

that the questionnaire could not capture place-specific types of information - for instance, sensory and modality-specific features - that are crucial when it comes to cognitive processing of familiar places. Secondly, as discussed in [2, 60] , the identity of personally familiar places may be harder to process than that of personally familiar people - for various reasons that could be evolutionary, [132] social, [133] or related to the availability of semantic features during retrieval [134]. This would then make it harder to correctly distinguish place-specific, as opposed to person-specific, signatures in

brain activity - a result that converges with the overall lower encoding scores for places reported here (Fig s 5 and Fig s 5) and that has previously been found in the literature [60, 134, 135] While we are unable to provide an answer to such questions with the current experiment, we believe this could be a fruitful direction for future research."

R1: "If the Introduction section is clearly written, the other sections need small revisions from yours:

- line 115: Fig 1 left portion  right portion

- line 116: We  we

- lines 116-117: .. had not either ... or  had neither .. nor"

Answer: Thanks for pointing out these typos, that we have now corrected.

R1: "- line 128: pleade add standard deviations as well"

Answer: We have now added all standard deviations.

R1: "- line132 and all ensuing occurrences: see Section  reference to Section is always missing"

Answer: Sorry for the mistake. This should now have been fixed.

R1: "- line 248: (Wikipedia or questionnairre)  why Wikipedia? in lines 155-157 I read that for individual entity you take names and a short description (occupations for people and type of places)"

Answer: Thanks for making us notice this point. We have now clarified when we used the texts (questionnaires or Wikipedia pages; to create the vectors) and when we employed categories (during the experiment).

NEW TEXT:

"For each famous individual entity we also collected the text from their Wikipedia page, under the assumption that such texts are a source of explicit knowledge regarding individual entities that can be mapped to the brain using their distributional information - an approach validated in neuroscience in [55] and in Natural Language Processing (NLP) in [41, 56, 57]. These texts will be used as described in Section 2.4.1 in order to extract semantic representations using language models. We also manually annotated for each famous person their occupation (e.g. politician, musician) and for each famous place its type (e.g. city, monument), since this information was used during the experimental task (see Section 2.2)"

R1: "- lines 251-254: here the syntactic structure is a bit odd and access to the content is quite difficult."

Answer: We have tried to clarify and simplify as much as possible the whole part dedicated to the vector extraction procedure.

NEW TEXT:

The rationale for this procedure is that words appearing in a text revolving around an entity carry a rich bundle of semantic information with respect to that entity. In other words we interpret the texts we collected as entity-specific distributional lexical information:

for personally familiar people and places, the answers to the questionnaires; for their famous counterparts, the text from their Wikipedia pages [41, 55, 56]. The procedure was matched across models, so as to avoid methodological confounds. For each individual entity, we split the text in passages of at least 20 words. We set a lower passage length threshold because, when encoding sentences for downstream use, LLMs have been shown to work better with rather long passages [71]. In order to ensure that text portions were long enough to work well with XLM, we rearranged the text so that sequential passages of at least 20 words were created (i.e. if a sentence were shorter than 20 words, we considered to be part of the same passage as the following sentence).

We chose 20 words as a threshold since in English sentence lengths are most commonly is between 10 and 30 words [72]).

After having encoded all passages of text using the language models, we retained only the vectors in the sequences corresponding to content words (i.e. open-class words: nouns, verbs, adjectives, adverbs) from the corresponding descriptive text (Wikipedia or questionnaire). We decided to follow [44] and exclude closed-class words such as

function words, since they do not carry semantic information. We reasoned that their presence would lead to the static language model being disadvantaged, since it has been shown that they struggle at representing function words properly [43, 73, 74].

Finally, we obtained a single entity representation in two steps. First, we averaged all the word vectors retained for each passage, thus obtaining one vector per passage.

Then, we averaged all of the vectors for the individual passages of texts [32, 55] [75] . Therefore, at the end of the procedure we had one vector representation for each individual entity per model, capturing the distributional lexical information contained in our small-scale textual data.

R1: "- line 274: at at (remove one of them)

- line 291: odd punctuation (sentence; and so on  sentence, and so on)

- lines 295-302: pay attention to the consecutio temporum."

Answer: Thanks - these should be fixed now.

Reviewer #2 (R2): 

R2: "Regrettably on reading the full paper I discovered that I am not competent to review it because there is virtually no linguistics in it but computational encephalography. I apologize for any inconvenience caused by my inept decision. Although I have recommended rejection it was a forced decision and should be ignored."

Answer: No worries, and thanks anyways for your help.

Reviewer #3 (R3): 

R3: "Thank you for letting me review this very interesting paper on how language models can encode subject-specific information. The topic itself is very interesting because usually language models try to capture concepts from large amounts of data, while in this case, we see how they can also be used to encode some fragments of personal life. The manuscript is well written and the study is properly evaluated."

Answer: Thanks for the insightful comments, suggestions and the positive feedback..

R3: "Specific comments:

Abstract:

- I suggest to rewrite the sentence "In terms of spatio-temporal clusters, ... (500-700ms)." for better clarity."

Answer: Thanks for the suggestion. We have now fully reformulated the sentence and modified the paragraph, hopefully making it clearer.

NEW TEXT:

"Using spatio-temporal EEG searchlight, we found that we could predict brain responses significantly better than chance earlier (200-500ms) in bilateral temporo-parietal electrodes and later (500-700ms) in frontal and posterior central electrodes."

R3: "Methods -> Famous entities:

- Lines 127-139: Have you checked that the assumptions of the t-test are met?"

Answer: We have now changed the analyses using nonparametric permutation tests, which make minimal assumptions about the underlying data distributions.

R3: "Methods -> Personally familiar entities:

- In line 154, you mention that you provided the text of the specific instructions given to the subjects in the code. Scrolling quickly through the repository I did not find it. I suggest that you 1) Update the Readme so that it is easier to find the information and 2) Provide a list of the questions/instructions as part of the manuscript (in the appendix?). I think it is a very interesting part of the study and that it should be included."

Answer: Really sorry for this - we forgot to update the repository. We have now added the translated questionnaire as Supplementary Information and we have updated the repository containing the data and the code online. 

R3: "Methods -> Spatio-temporal searchlight:

- I suggest to rewrite the sentence in lines 371-373: "Results are intepreted as above... at a given point in time" for better clarity."

Answer: Thanks for pointing this out. We have now reformulated the sentence, with the aim of making it easier to understand.

NEW TEXT:

We were also interested in going beyond patterns of activity widely distributed across the whole scalp, and look at specific areas on the scalp where a model could explain evoked reponses. To this aim, we implemented a searchlight encoding analysis.

Searchlight allows to find in a bottom-up fashion localized clusters of brain activity associated with an experimental condition, while exploiting the high sensitivity of multivariate analyses [91, 92]. In practice, searchlight consists of running the encoding analyses repeatedly across smaller clusters of electrodes on the scalp. To reduce the computational effort , we followed previous work [93] and used spatio-temporal clusters, where multiple time points are considered for each electrode within the cluster. As in [94], we employed a temporal radius of 50ms and a spatial radius of 30mm (i.e. a cluster contains evoked activity for 100ms, for electrodes falling within a circle having a diameter of 60mm). We computed statistical significance tests using the TFCE method described in Section 2.5.6. If the p-value for a cluster of electrodes at a given point in time fell below 0.05, we considered encoding to be significantly above chance.

R3: "Methods -> Statistical testing:

- Line 380: We ran one-tailed statistical tests: which ones?"

Answer: Thanks for noticing this - we have now specified which statistical tests were used..

R3: "Results:

- It would be interesting to see the same figure as Figure 4, but for word2vec, so that it would also be possible to compare how this model performed for the different tasks."

Answer: Thanks for the suggestion. We have now added the full figure for word2vec (now Figure 5), and added the relevant parts to the Results section - which indeed makes clearer where the difference in performance between the two models stems from.

NEW FIGURE:

R3: "- Figure 3, Figure 4: it would be nice to add that the time radius for the searchlight was 50ms.

- Figure 3, Figure 4: it would also be nice to add the x-axis under the searchlight coding."

Answer: Thanks for both suggestions - we have now added the temporal radius in the figure, and the x-axis under the searchlight encoding results..

R3: "- Figure 4: What do the points indicated as stars in the graph represent?"

Answer: Thanks a lot for spotting this! We had wrongly used a previous version of the figure, where stars represented points approaching significance ($p<.08$). Now only statistically significant ($p<0.05$) points are marked with a dot.

R3: "Other comments:

- The hyperlinking of the sections did not work, see lines 132, 158, 178, 182, 240, 284, 497, 510,...

- Line 204: "Each trial proceeded as followS". Missing S.

- The acronym MNE is never spelled out.

- Line 285: "pooling for the top four layerS". Missing S.

- Figure 3 and Figure 4: space missing between "scalp locations" and "(searchlight)"."

Answer: Thanks for spotting these typos - they are now corrected.

Reviewer #4 (R4): 

R2: "In this paper, the authors use language models(both small and big model) to build a connection between semantic knowledge from personally familiar individual entities and semantic information in brain. And a series experiment show that encoding performance was significant in a large time window (200-800ms).I think the research topic is very interesting and owns practical value. " 

Answer: Thanks for your feedback and the valuable comments.

R4: "There are some questions as follows, (1) why you choose 10 words to to form a topic?"

Answer: Thanks for the suggestion. We have now explicitly indicated from where we took each experimental parameter from.

NEW TEXT:

"We pre-trained a word2vec model o

---

## [Decision Letter · Decision Letter 1]

17 Sep 2024

Family lexicon: using language models to encode memories of personally familiar and famous people and places in the brain

PONE-D-23-26777R1

Dear Dr. Bruera,

We’re pleased to inform you that your manuscript has been judged scientifically suitable for publication and will be formally accepted for publication once it meets all outstanding technical requirements.

Kind regards,

Michal Ptaszynski, PhD

Academic Editor

PLOS ONE

Additional Editor Comments (optional):

Reviewers' comments:

Reviewer's Responses to Questions

**Comments to the Author**

1. If the authors have adequately addressed your comments raised in a previous round of review and you feel that this manuscript is now acceptable for publication, you may indicate that here to bypass the “Comments to the Author” section, enter your conflict of interest statement in the “Confidential to Editor” section, and submit your "Accept" recommendation.

Reviewer #1: All comments have been addressed

2. Is the manuscript technically sound, and do the data support the conclusions?

Reviewer #1: Yes

3. Has the statistical analysis been performed appropriately and rigorously? 

Reviewer #1: Yes

4. Have the authors made all data underlying the findings in their manuscript fully available?

Reviewer #1: Yes

5. Is the manuscript presented in an intelligible fashion and written in standard English?

Reviewer #1: Yes

6. Review Comments to the Author

Reviewer #1: I strongly appreciate that the authors took into consideration all detailed comments from the reviewers, thus providing a revised version of their work which is highly relevant to a wide research community.

Some minorities should be addressed:

Line 279: there is a “is” that must be cancelled.

Line 676: Fig s 3 and 5

Line 721: Fig s 5 and Fig s 5

7. PLOS authors have the option to publish the peer review history of their article (what does this mean?). If published, this will include your full peer review and any attached files.

Reviewer #1: **Yes: **Claudia Marzi

---

## [Editor Report · Acceptance letter]

2 Oct 2024

PONE-D-23-26777R1 

PLOS ONE

Dear Dr. Bruera, 

I'm pleased to inform you that your manuscript has been deemed suitable for publication in PLOS ONE. Congratulations! Your manuscript is now being handed over to our production team.

Kind regards, 

on behalf of

Dr. Michal Ptaszynski 

Academic Editor

PLOS ONE